

# Assessment of college students' mental health status based on temporal perception and hybrid clustering algorithm under the impact of public health events

Mao Li[1] and Fanfan Li[2]

[1] Sichuan Vocational College of Health and Rehabilitation, Zigong, Sichuan, China
[2] Student Affairs Department, Huanggang Normal University, Huanggang, Hubei, China

## ABSTRACT

The dynamic landscape of public health occurrences presents a formidable challenge to the emotional well-being of college students, necessitating a precise appraisal of their mental health (MH) status. A pivotal metric in this realm is the Mental Health Assessment Index, a prevalent gauge utilized to ascertain an individual's psychological well-being. However, prevailing indices predominantly stem from a physical vantage point, neglecting the intricate psychological dimensions. In pursuit of a judicious evaluation of college students' mental health within the crucible of public health vicissitudes, we have pioneered an innovative metric, underscored by temporal perception, in concert with a hybrid clustering algorithm. This augmentation stands poised to enrich the extant psychological assessment index framework. Our approach hinges on the transmutation of temporal perception into a quantifiable measure, harmoniously interwoven with established evaluative metrics, thereby forging a novel composite evaluation metric. This composite metric serves as the fulcrum upon which we have conceived a pioneering clustering algorithm, seamlessly fusing the fireworks algorithm with K-means clustering. The strategic integration of the fireworks algorithm addresses a noteworthy vulnerability inherent to K-means—its susceptibility to converging onto local optima. Empirical validation of our paradigm attests to its efficacy. The proposed hybrid clustering algorithm aptly captures the dynamic nuances characterizing college students' mental health trajectories. Across diverse assessment stages, our model consistently attains an accuracy threshold surpassing 90%, thus outshining existing evaluation techniques in both precision and simplicity. In summation, this innovative amalgamation presents a formidable stride toward an augmented understanding of college students' mental well-being during times of fluctuating public health dynamics.

Corresponding author
Fanfan Li, lffxg@hgnu.edu.cn

# INTRODUCTION

Public health crises, exemplified by the COVID-19 pandemic, exert profound psychological ramifications. It is noteworthy that a majority of afflicted individuals evinced psychological

perturbations, encompassing concerns related to fundamental sustenance, apprehensions surrounding personal and familial well-being, heightened states of anxiety, and disruptions in sleep patterns. Scholarly investigations have diligently scrutinized the impact of COVID-19 upon various strata, encompassing frontline medical personnel, individuals afflicted by the virus, and the broader populace (*Son et al., 2020*; *Deckro et al., 2002*; *Wang et al., 2018*). Conspicuously, both confirmed and presumptive COVID-19 patients have exhibited varying degrees of post-traumatic stress symptoms, with the intensity of these manifestations inversely correlated to the quality of sleep attained; furthermore, a subset of patients may be susceptible to the development of post-traumatic stress disorder (PTSD).

Within this complex tapestry, a distinctive cohort emerges in the form of college students, whose lives and academic pursuits have been uniquely affected by the strategies employed for epidemic containment. The COVID-19 crisis has imparted a disruptive cadence upon the lives and academic trajectories of these scholars. Notably, it has engendered a milieu of unease, despondency, and trepidation amongst university students, potentially imperiling their mental well-being, unless they are proffered adequate educational guidance and recalibration. This underscores the exigency of furnishing educational counseling and adaptational interventions to mitigate the deleterious psychological consequences precipitated by the pandemic (*Browning et al., 2021*; *Li & Liang, 2022*). In times of public health crises, the mental well-being of college students takes on heightened significance. The impact of such crises extends beyond physical health, encompassing emotional, psychological, and academic realms. Understanding and assessing the mental well-being of these students is crucial, as it directly influences their ability to cope, learn, and succeed. By recognizing the profound interplay between mental health and academic performance during public health challenges, institutions can proactively offer support and resources to foster resilience and ensure students' overall well-being.

The inadequate effect of psychological prediction has always been a source of concern for researchers and experts. Using machine learning, it is possible to optimize and adapt the algorithm's parameters to produce the optimal model, *i.e.,* the performance of the machine learning model can be altered while the data remains unchanged (*Wang et al., 2020*; *Zulkefly & Baharudin, 2010*; *Harren, 1979*). Several researchers predicted the efficacy of cognitive behavioral therapy for children with obsessive-compulsive disorder using four distinct machine learning algorithms and a multivariate logistic regression model (*Garlow et al., 2008*). The results demonstrated that the machine learning system predicted the treatment effect with an average accuracy of approximately 80%. Other researchers have used clinical and demographic data from patients with borderline personality disorder, in conjunction with multimodal magnetic resonance imaging and random forest models, to accurately predict treatment effects for borderline personality disorder with a sensitivity of over 70% (*Pritchard & Wilson, 2003*; *Liu, 2021*; *Means, Lichstein & Epperson, 2000*).

Several researchers have implemented machine learning into the field of psychological medical diagnosis. The MH status of college students has been evaluated and studied by a number of scholars using data mining theory. Some researchers employed cluster analysis methods, two-step clustering and quick clustering, to cluster adolescent risk behaviors and develop intervention programs for adolescents (*Talasbek et al., 2020*). In

other investigations, the technique for median clustering was utilized to cluster maternal MH care.

In time psychology, time perception refers to the continuous and sequential reaction of humans to time stimuli operating directly on their own senses. That is, humans can evaluate the perception of time duration and velocity without the use of a timer (*Meck, 1996*). The sense of time is comparable to the perception of other elements, including color, form, and temperature. It is the human species' natural perceptual instinct. Incorporating time perception into the psychological evaluation of college students can thereby enhance the evaluation index system (*Coelho et al., 2004*).

In the realm of scholarly exploration, it is notable that the preponderance of extant investigations employ supervised machine learning methodologies to cultivate models, harnessing meticulously annotated data samples for model acquisition. Nevertheless, within the domain of mental health (MH), a conspicuous lacuna is evident, primarily manifesting as a multitude of nascent MH conditions that confront a dearth of lucid characterization or exhibit an ephemeral and nebulous disposition. In response, this scholarly discourse introduces the domain of time perception from the annals of psychology, seamlessly integrating it into the construction of an evaluative framework. This novel paradigm serves as an avenue to address the inherent deficiencies, facilitating the augmentation of evaluation accuracy. Moreover, it assumes a pivotal role in elucidating the nuanced MH phases experienced by college students, thereby presenting a compass for informed psychological counsel and timely intervention.

Central to this approach is the translation of temporal perception into a quantifiable metric, ingeniously amalgamated with established evaluative metrics to engender a unified metric. This novel metric lays the foundation for our ensuing proposition: a pioneering clustering algorithm underpinned by the amalgamated metric. This innovative clustering algorithm, a confluence of the fireworks algorithm and K-means, is meticulously designed to surmount a notable challenge intrinsic to the K-means algorithm—its susceptibility to converging to local optima.

The fireworks algorithm is strategically integrated to address and surmount this challenge, imparting resilience and expanding the algorithm's potential to unearth global optima. By embracing this synergistic amalgamation, our endeavor is to cultivate a transformative methodology that transcends the prevailing limitations, proffering heightened precision in evaluation while fostering a deeper understanding of the intricate MH trajectories traversed by college students. Ultimately, this approach stands poised as an invaluable compass for psychological guidance and efficacious interventions, distinctly illuminating the pathway toward ameliorated mental well-being within the academic fraternity.

## RELATED WORK

### Evaluation classification algorithm

Cluster analysis is an essential component of data mining technology. By separating the data set into numerous classes, comparable samples are classified into one class based on

the properties of the data, while different samples are sorted into separate classes to ensure homogeneity within classes and heterogeneity between classes (*Meck, 1996*). Scholars from around the world have incorporated cluster analysis into psychological prediction and evaluation, using various clustering algorithms to analyze psychological indicators in order to provide guidance for psychological intervention in advance, as a result of the widespread application of data mining technology (*Coelho et al., 2004*; *Ariff, Bakar & Zamzuri, 2020*).

K-means clustering is a partitioning-based technique with low time complexity, high clustering efficiency, and excellent clustering quality. Some academics in the industrial sector choose the K-means algorithm and the hierarchical clustering algorithm to classify temperature fluctuations. Some researchers developed self-organizing feature mapping (SOM) based on the K-means method (*Wu, Zhao & Guo, 2020*; *Chang & Yang, 2009*; *Lee & Macqueen, 1980*; *Qin & Gui, 2022*). They employed a SOM training data set and K-means clustering based on the output findings of the training set to generate superior clustering results, which led to the visualization and understanding of the model being successful (*Beauchaine & Beauchaine III, 2002*).

The K-means algorithm is also sensitive to the starting center and requires data distribution, which are both downsides. Hierarchical clustering techniques typically have a high temporal complexity, and popular algorithms such as ROCK and Chameleon do not support big data sets. SOM is a model-based clustering approach with shortcomings including high temporal complexity, inability to handle huge data sets, and clustering results that are sensitive to model parameters. The model's advantage comes in the fact that it can accurately describe the data (*Li et al., 2021*; *Jamali & Ayatollahi, 2015*; *Tang, 2021*). As a result of a comprehensive examination of the clustering algorithm, discriminant analysis, principal component analysis, and other techniques have been utilized to interpret clustering results. Using principal component analysis, a number of scholars have tackled the problem of indicators' high repetition. In addition, the results of the study indicate that discriminant analysis and principal component analysis have yielded fruitful interpretations of clustering results. The TwoStep algorithm is an enhanced hierarchical clustering algorithm that decreases the algorithm's temporal complexity, automatically determines the ideal cluster number, and scales well (*Dong & Shen, 2022*).

State assessment is an essential component of daily management. It is a sophisticated and abstract nonlinear issue that is affected by numerous variables and has a complex change rule. Due to the inability of the traditional state assessment model to fulfill the requirements of the present complicated quality assessment task, the application of intelligent algorithms to increase the efficiency and accuracy of quality assessment has become the focus of current research.

Some researchers presented a state assessment technique based on the adaptive BPNN model, which introduced the adaptive learning rate momentum to optimize the model's network topology and ensure its stability. Some academics have proposed a research methodology for project evaluation based on the analytic hierarchy approach (AHP) (*Nelsen, Kayaalp & Page, 2021*; *Irawan, 2019*). Integrating multi-person and multi-attribute aspects, qualitative analysis and quantitative computation are used to assess the quality of management. Some researchers have proposed a design scheme for an association

mining-based foreign language evaluation model that combines the data analysis method with the quantitative language evaluation model, extracts the evaluated association rule features from the reconstructed phase space as the clustering center of information fusion, and realizes the optimal design of the evaluation model *via* adaptive regression analysis. Using the AHP, some researchers designed a method for providing a realistic assessment of the quality of network resources, created assessment objectives-centered material, and established a hierarchical structure model (*Sorour et al., 2014*; *Yanfang & Yamin, 2021*).

The Fireworks Algorithm (FWA) is a novel form of swarm optimization technique that simulates the search process of neighborhood space during the explosion of fireworks. It is capable of balancing global and local search, and has the benefits of simple implementation, straightforward operation, and powerful search capabilities (*Jiang et al., 2022*; *Li & Tan, 2019*; *Xue, 2020*). The Fireworks Algorithm, inspired by the explosion of fireworks in the night sky, has found promising applications in optimization problems. This nature-inspired metaheuristic approach simulates the explosion of fireworks to enhance the search process within complex solution spaces. Through its utilization, the algorithm has exhibited remarkable efficacy in solving a variety of optimization problems, ranging from engineering design and parameter tuning to financial portfolio optimization (*Karimov & Ozbayoglu, 2015*; *Rahmani et al., 2015*).

On the other hand, the K-means algorithm, a fundamental clustering technique, has established its dominance in partitioning datasets into distinct groups. By iteratively assigning data points to clusters and recalculating cluster centroids, the algorithm minimizes the within-cluster variance, effectively grouping similar data points together. Widely used in data analysis and pattern recognition, the K-means algorithm has proven instrumental in applications such as image segmentation, customer segmentation, and anomaly detection.

Drawing parallels between the two, it becomes evident that while the Fireworks Algorithm excels in optimization tasks by efficiently exploring solution spaces, the K-means algorithm specializes in data clustering by identifying inherent patterns within datasets. Interestingly, their areas of application are not mutually exclusive. Recent research has begun to explore the synergy between these algorithms, leveraging the Fireworks Algorithm's global exploration capabilities to initialize K-means and enhance its convergence towards better clustering solutions. This fusion of methodologies showcases the dynamic nature of algorithmic development, where distinct techniques can complement each other to provide novel and improved solutions to complex problems.

In conclusion, the Fireworks Algorithm and the K-means algorithm, though operating in different domains, exhibit remarkable potential in their respective application areas. Their unique characteristics and strengths offer opportunities for cross-disciplinary utilization, opening doors to innovative hybrid approaches that harness the best of both worlds. As the field of optimization and clustering continues to evolve, these algorithms stand as notable pillars, shaping the landscape of intelligent problem-solving techniques.

## METHOD

### Setting of temporal perception

Translating temporal awareness into quantifiable metrics is a crucial step in enhancing comprehension of methodology, particularly in fields that involve time-dependent processes or data analysis. This process involves converting the inherent understanding of time-related factors into measurable and meaningful indicators. this include: (1) Identify key temporal factors, which includes variables such as time intervals, durations, sequencing of events, frequency of occurrences, timestamps, and historical trends. if user are analyzing website user behavior, they might define metrics such as "average time spent on page," "time between interactions," "conversion rate over time," or "rate of content consumption." (2) Temporal aggregation: users may need to aggregate the temporal data to create meaningful intervals or time units for analysis. This could involve grouping data into hours, days, weeks, months, or other relevant timeframes. (3) Validation and iteration: Validate the translated metrics by comparing them with existing benchmarks, theories, or expectations. If necessary, refine metrics and methods based on feedback and insights gained from the analysis.

### FWA-K-means

K-means algorithm can be optimized by using FWA with the ability of balancing global and local search, and the obtained data results can be used as the initial clustering center of K-means algorithm to solve the problem that K-means algorithm is easy to fall into local optimum. This is done in an effort to address the issues of low accuracy, poor reliability, and low efficiency of the mental state assessment model for college students.

The steps of the proposed hybrid clustering method are as follows:

Step 1: Initialize parameters by entering the fireworks scale n, current iteration number t and maximum iteration number T.

Step 2: Calculate the fitness, explosion radius, and number of sparks generated by each firework.

In FWA, there are two parameters that play a decisive role. The first is the explosion radius $r_i$ of firework i

$$r_i = \frac{f - f_{\min}}{\sum_{i=1}^{n}(f - f_{\min})}\bar{r} \qquad (1)$$

where $f$ is the fitness function, and $\bar{r}$ is

$$\bar{r} = \frac{\sum_{i=1}^{n} r_i}{n} \qquad (2)$$

The second key parameter is the number of sparks $s_i$

$$s_i = \frac{f_{\max} - f}{\sum_{i=1}^{n}(f_{\max} - f)}k \qquad (3)$$

where $k$ is a constant, and when $s_i \leq aW$, $s_i = [0.1W]$, when $s_i \leq 0.5W$, $s_i = [0.5W]$

Step 3: Calculate the fitness of spark and the fitness of Gauss mutation spark.

The formation of a spark is shown as

$$x'_{ij} = x_{ij} + l_i U[-1, 1] \tag{4}$$

where $j$ is the dimension of the spark, $l_i$ is the uniformly distributed random number, and $U[-1, 1]$ is the scaling parameter.

Step 4: Select n individuals from the fireworks, sparks, and Gauss variant sparks as the next generation fireworks.

The mutation operator is introduced in FWA to generate Gaussian sparks and increase the diversity of the population.

$$x''_{ij} = x_{ij} N(0, 1) \tag{5}$$

where $N(0, 1)$ isa standard normal distribution

Step 5: Repeat Step 2–4.

Step 6: Take N fireworks individuals as the initial cluster center point, calculate the cluster center distance, and divide each data object into the nearest cluster.

The calculated value of the distance between samples is

$$d(x_i, x_j) = \sqrt{\sum_{i=1}^{n} (x_i - x_j)^2} \tag{6}$$

The update of the cluster center is shown as follows

$$c_i = \frac{\sum_{x_i \in c_i} x_i}{|c_i|} \tag{7}$$

It is necessary for the K-means clustering method to repeatedly update the divided categories as well as the clustering center point c. up to the point where the termination condition is satisfied. The number of iterations must reach a maximum before the default termination condition is met, which is either the objective function of the algorithm falling below the threshold or the number of iterations reaching that maximum.

The loss function is

$$L = \sum_{i=1}^{n} \sum_{x_i \in c_i} |x_i - c_i|^2 \tag{8}$$

Step 7: Update the clustering center of each category.

The flow chart of the hybrid clustering algorithm is shown in Fig. 1.

## EXPERIMENTS AND ANALYSIS

### Establishment of evaluation system

The design of the intelligent assessment index system of college students' MH is the initial step in the construction of the evaluation model of college students' MH. In time psychology, time perception refers to the continuous and sequential response of humans to time stimuli affecting their own senses. Thus, humans can assess the perception of time duration and velocity without a timer. Comparable to the perception of other

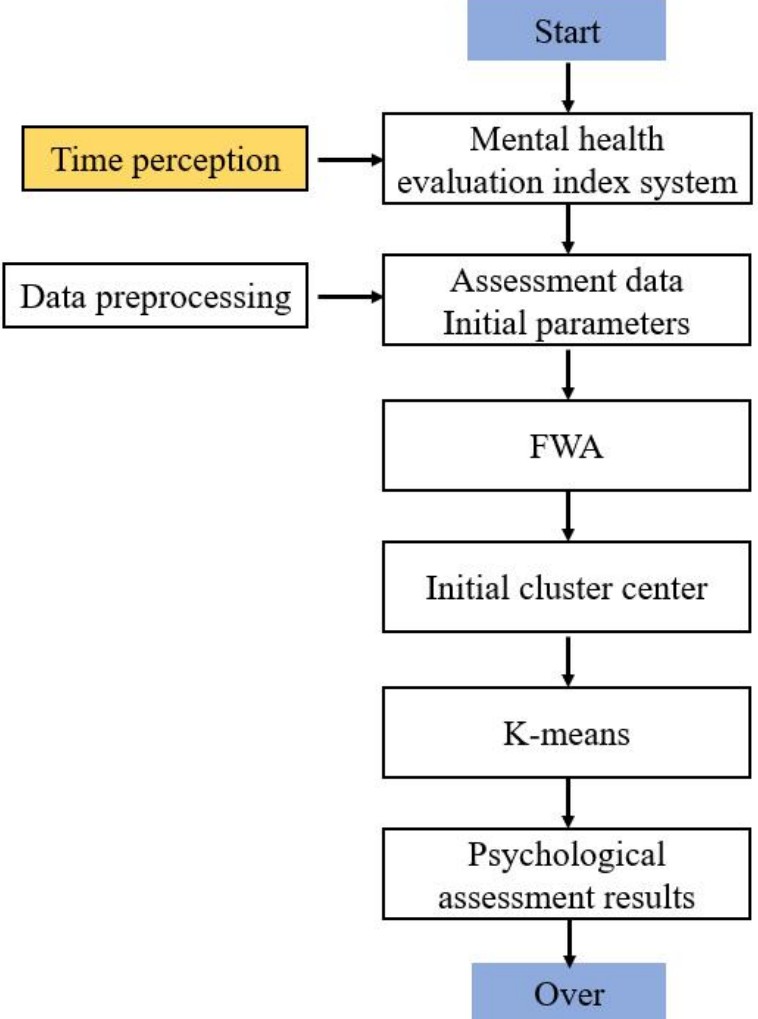

**Figure 1   Flow chart of hybrid clustering algorithm.**

elements, such as color, shape, and temperature, is the perception of time. It is the natural perceptual instinct of the human species. By including time perception into the psychological evaluation of pupils, the evaluation index system can be improved. Combining the current scenario of college students' MH and the notion of time perception in psychology, Fig. 2 depicts the intelligent evaluation index system for college students' MH.

Further, the index is quantified. To keep it simple and universal, each indicator is divided into three levels, with scores ranging from 0 to 10. A score greater than or equal to 8 indicates normal, a score ranging from 3 to 7 indicates average, and a score less than 3 indicates certain obstacles in this area.

For the final MH score, referring to the previous literature, this article divided it into four grades, namely normal, mild anxiety, moderate anxiety and severe anxiety.

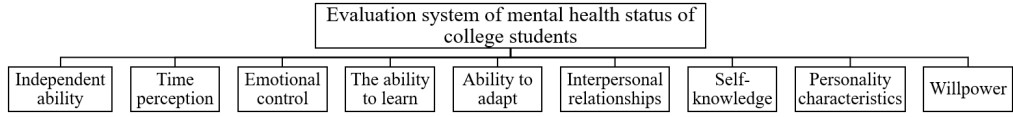

**Figure 2  The evaluation index system for college students' MH.** For the final MH score, referring to the previous literature, this article divided it into four grades, namely normal, mild anxiety, moderate anxiety and severe anxiety.

**Table 1  Hyperparameter settings.**

| Hyperparameter | Value |
|---|---|
| Maximum number of iterations | 100 |
| Individual variation | 6 |
| The explosion radius | 25 |
| The fireworks scale | 6 |
| The clustering center | 10 |

To validate the proposed hybrid clustering algorithm for the effectiveness of college students MH evaluation, select six college students as the research object, because the data is difficult to obtain, in this article, the number of each college selection for 200 people, first by the psychological association of colleges and universities to makes a preliminary test of the number of selected college students psychological rating, namely the sample label.

In order to verify the effectiveness of the proposed model, simulation experiments are carried out. The experimental simulation environment is Intel I9-10900K, the memory is 64GB, and the GPU is RTX3080.

The hyperparameter Settings are shown in Table 1

## Model comparision

In this research, the assessment results of the hybrid clustering model are compared with K-means, K-means ++, and K-means+SOM to demonstrate its advantages. K-means++ is an improved initialization technique for the K-means clustering algorithm. The primary goal of K-means++ is to select initial cluster centroids in a way that enhances the convergence speed and quality of the final clustering results. The traditional K-means algorithm often suffers from sensitivity to the initial placement of centroids, which can lead to suboptimal clustering outcomes or slow convergence.

K-means++ addresses this issue by following a systematic approach to initialize the centroids. The algorithm selects the first centroid randomly from the data points. Subsequent centroids are then chosen with a higher probability of being far from existing centroids, effectively spreading out the initial centroids. This initialization process helps K-means converge to a better solution and often requires fewer iterations to achieve convergence compared to random initialization.

K-means+SOM refers to a combination of the K-means clustering algorithm and the Self-Organizing Map (SOM), which is a type of artificial neural network used for data
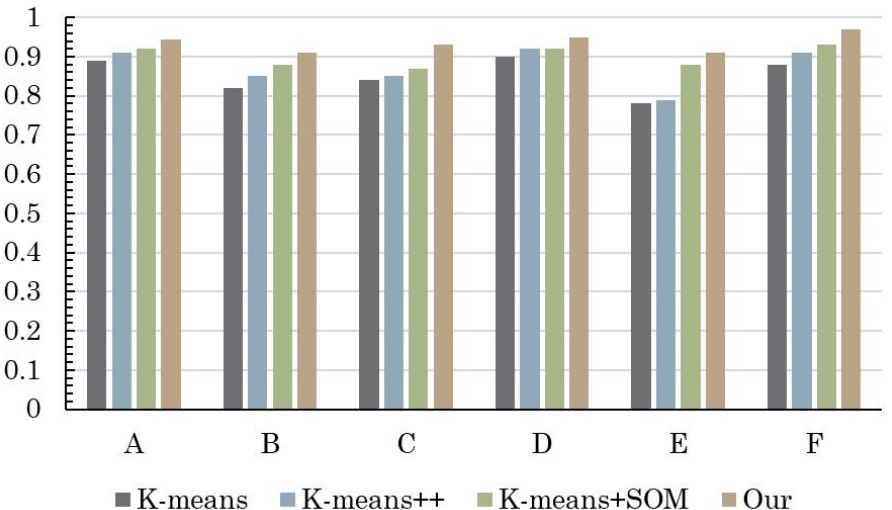

**Figure 3** Comparison of clustering accuracy of different methods.

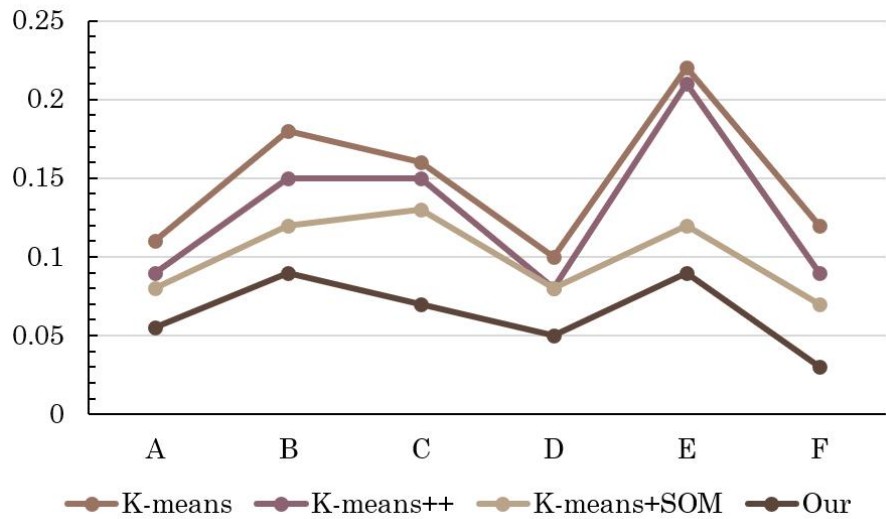

**Figure 4** Comparison of error rate of different methods.

visualization and clustering. In this hybrid approach, the strengths of both K-means and SOM are leveraged to improve clustering results.

The insightful comparison outcomes are visually depicted through the intricate graphical representations found in Figs. 3 and 4. Figure 3 elucidates the evaluation accuracy achieved by a diverse array of models, offering a glimpse into their respective strengths and weaknesses. Meanwhile, Fig. 4 delves deeper into the intricacies of error rates, shedding light on the nuances of the evaluation models' performance.

Remarkably, the results reveal a clear hierarchy in performance, with the conventional K-means algorithm displaying the least favorable accuracy, closely trailed by the enhanced

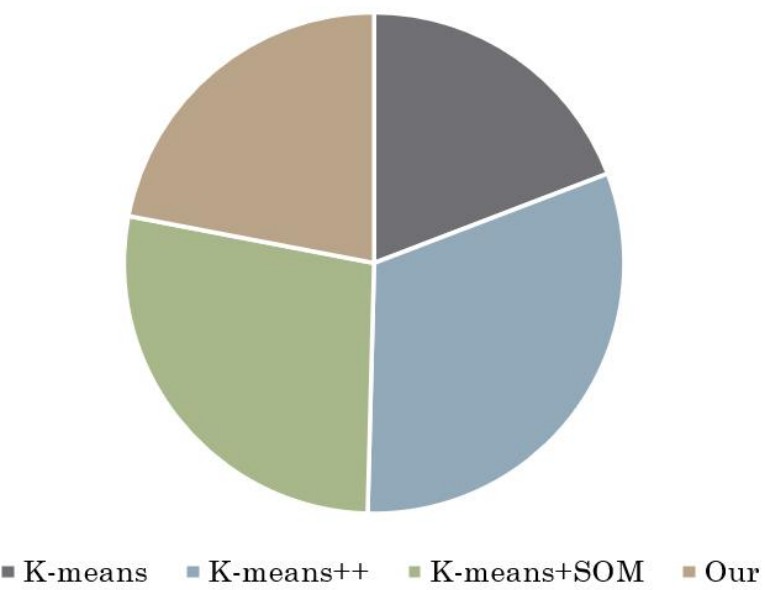

**Figure 5** **Comparison of modeling time of different algorithms in University A.** The modeling time of different models is compared and two universities are randomly selected, as shown here and in Fig. 6.

K-means++. However, the real breakthrough comes with the innovative integration of K-means+SOM, where the fusion of the SOM algorithm demonstrates its potential in substantially augmenting clustering accuracy. This notable improvement is accompanied by a significant reduction in the error rate, underscoring the efficacy of the hybrid approach.

The pivotal significance of the method detailed in this article becomes even more pronounced upon scrutinizing its impact on student MH (mental health) status evaluation. The discernible increase in precision signifies a more nuanced understanding of students' psychological well-being. The reduced error rate in assessing students' MH status implies a higher level of confidence in the evaluation process, which in turn contributes to the overall improvement of students' psychological health status assessment.

Moreover, an additional facet of this study involves a comprehensive examination of the modeling time required by various models. To provide a more robust assessment, two universities were randomly selected for testing, yielding illustrative insights displayed in Figs. 5 and 6. Notably, these figures shed light on the intricate interplay between modeling time and algorithmic intricacies.

Upon careful observation, a noteworthy revelation surfaces –the hybrid clustering algorithm introduced in this article, despite integrating the fireworks algorithm, remains judiciously efficient in its modeling time. In sharp contrast to other algorithms under scrutiny, the hybrid approach manages to strike a harmonious balance. It seamlessly incorporates the prowess of the fireworks algorithm while ensuring that the modeling time does not experience a significant escalation when juxtaposed against the established benchmark set by the K-means algorithm.

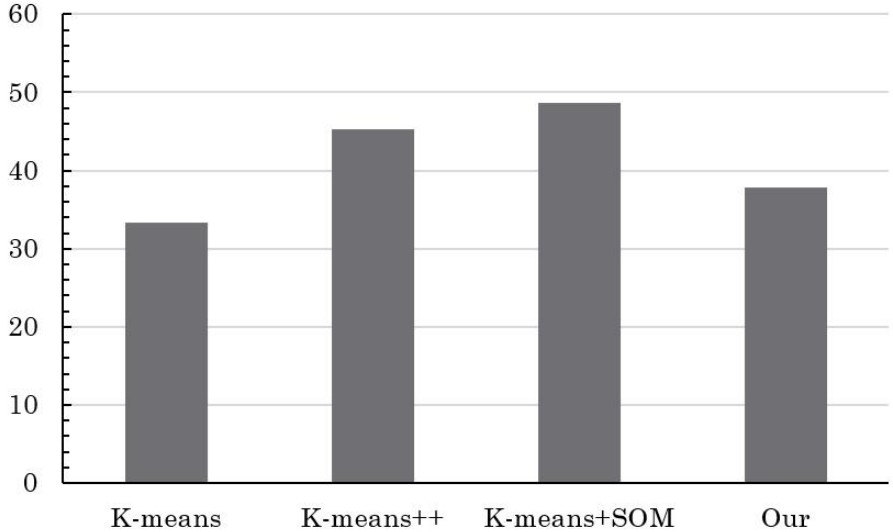

**Figure 6** **Comparison of modeling time of different algorithms in University D.** The modeling time of different models is compared and two universities were randomly selected for testing, as shown in Figs. 5 and this figure.

This facet of the study underscores a pivotal achievement—the successful integration of advanced algorithmic techniques without incurring a notable trade-off in terms of computational efficiency. This substantiates the algorithm's practical applicability and reinforces its potential for real-world implementation. The ability to optimize clustering accuracy without imposing an undue burden on computational resources adds a layer of practicality that enhances the overall appeal and feasibility of the proposed approach. In essence, the analysis of modeling time, as depicted in Figs. 5 and 6, contributes a holistic perspective to the evaluation of the hybrid clustering algorithm's efficacy. The findings underscore not only its substantial accuracy enhancements but also its commendable computational efficiency, further solidifying its status as a promising contender for enhancing diverse evaluation processes.

## DISCUSSION

Comparing the results against the backdrop of existing research, the validity of the hybrid clustering algorithm takes center stage. This algorithm, incorporating the FWA (Fireworks Algorithm) known for its prowess in balanced global and local searches, presents a unique and effective solution. Its synergy with the K-means clustering technique results in the construction of a high-precision curriculum quality evaluation model. This distinctive model demonstrates the potential to revolutionize the evaluation landscape, offering a more accurate and comprehensive approach to curriculum quality assessment.

In conclusion, the comprehensive analysis of the comparison outcomes and the subsequent discussions serve to underscore the significance of the methodology proposed in this study. The findings not only shed light on the limitations of traditional approaches

but also emphasize the potential for innovation through the fusion of algorithms, as showcased by the successful integration of K-means+SOM and FWA. This study opens up new avenues for enhancing evaluation accuracy and effectiveness, particularly in the realm of student psychological well-being and curriculum quality assessment.

Moreover, under the emergency of public health events in Beijing, colleges and universities should, on the one hand, actively improve the economic support system, provide adequate psychological counseling services, conduct rich psychological education activities, and take other measures to improve the MH level of students from low-income families and prevent the formation of a strong inferiority complex. On the other hand, the MH of college students is strongly supported by their families. On the path of development for college students, if parents can adopt scientific and effective educational concepts, enhance parent–child contact, and provide enough spiritual and material support, it will be of tremendous assistance in maintaining their MH. Therefore, it is advised that relevant departments expedite the creation and promulgation of relevant regulations to promote the MH level of college students, in order to train for the country comprehensive talents with high professional quality and healthy psychological quality.

## CONCLUSIONS

In order to accurately determine the MH condition of college students when they are under the influence of public health events, we devised a method based on temporal perception and a hybrid clustering algorithm to assess the MH status of college students. This method was utilized to assess college students' MH. This study makes a novel addition to the growth of the psychological evaluation index system by innovating the incorporation of temporal perception in spatio-temporal psychology during the development of the MH judgment index system. In addition, a hybrid clustering method that combines the fireworks algorithm and the K-means algorithm is used to address the issue that the K-means algorithm quickly falls into a local optimum, and then the MH of college students is evaluated. Experiments have demonstrated that the developed hybrid clustering method accurately captures the changing MH features of college students. In addition, this approach is both simpler and more accurate than the current evaluation methods.

## ACKNOWLEDGEMENTS

We would like to thank the anonymous reviewers whose comments and suggestions helped improve this manuscript.

### Funding
The authors received no specific funding for this study.

### Competing Interests
The authors declare there are no competing interests.

## Author Contributions

- Mao Li conceived and designed the experiments, analyzed the data, prepared figures and/or tables, and approved the final draft.
- Fanfan Li performed the experiments, performed the computation work, authored or reviewed drafts of the article, and approved the final draft.

## Data Availability

The dataset is available at Zenodo: Lischer, Suzanne, Safi, Netkey, & Dickson, Cheryl. (2021). Remote learning and students' mental health during the Covid-19 pandemic: Dataset and questionnaire [Data set]. Zenodo. https://doi.org/10.5281/zenodo.5708255.

## Supplemental Information

Supplemental information for this article can be found online at http://dx.doi.org/10.7717/peerj-cs.1586#supplemental-information.

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
