# Peer review of "Assessment of college students’ mental health status based on temporal perception and hybrid clustering algorithm under the impact of public health events"

_PeerJ Computer Science, doi:10.7717/peerj-cs.1586_

## Round 0.1 · original submission · Major Revisions

Both reviewers have provided feedback suggesting several enhancements to the manuscript. Such as the authors should succinctly summarize key experiment findings related to college students' mental well-being shifts and substantiate claims about the hybrid clustering algorithm's superiority using concrete evidence.

Additionally, improvements in the literature section are needed, such as better commentary between the Fireworks Algorithm and K-means algorithm's application status. The research objectives must be explicitly stated, emphasizing contributions to mental health assessment knowledge. The authors should stress adopting a holistic psychological viewpoint, outlining how new indicators enrich the existing system and overcome past limitations.

A concise overview of the Fireworks Algorithm and its advantages should be provided.

Lastly, in conclusion, the practical significance and applications of findings in college students' mental health assessment should be discussed. Please see the reviewers' comments in detail. Thank you!

Reviewer 1 ·

Basic reporting

This study proficiently determined the psychological well-being of university students amidst the impact of public health incidents. Additionally, it introduced an assessment approach for gauging the mental health status of college students, leveraging a hybrid clustering algorithm that incorporates time-awareness. This research demonstrates ingenuity by integrating the notion of time perception into the development of a comprehensive index system for mental health evaluation, thus contributing to its further advancement.
This paper is innovative and the writing logic is rigorous.

Experimental design

The following are specific suggestions for modification to improve the quality of the paper.


Consider providing more coherent and succinct introductions that emphasize the importance of assessing the mental well-being of college students impacted by public health crises.

Enumerate specific public health events or instances relevant to the study to enhance the context and applicability of the research.

Emphasize the innovative and unique nature of the proposed metrics derived from time-aware and hybrid clustering algorithms.

Elaborate on the process of translating temporal awareness into quantifiable metrics to enhance comprehension of the methodology.

The article's organization structure and title should be revised to establish a more substantial logical connection among the various sections.

Precisely define "k-means++" and "K-means+SOM" to ensure clarity.

Validity of the findings

It would be preferable for the authors to summarize the key findings of the experiment, specifically in terms of accurately characterizing shifts in the mental well-being of college students.

Support the claim that the proposed hybrid clustering algorithm is simpler and more accurate than existing evaluation techniques by providing concrete evidence or statistical results.

Including references is crucial as it allows readers to understand the research's foundation and the previously cited works. Authors should review the format and specifications of the references.

Additional comments

see above comments

Reviewer 2 ·

Basic reporting

The author employs a hybrid clustering approach that combines the fireworks algorithm
with the K-means algorithm to address the issue of the K-means algorithm's
susceptibility to local optima. Subsequently, the mental health of college students is
evaluated using this method. The proposed hybrid clustering method effectively
captures the dynamics of psychological traits in college students.
The following suggestions are provided to enhance the quality of research justification,
contributions, originality, and findings:
1. Please ensure the use of precise and technical language throughout the abstract,
avoiding ambiguity or overly general statements.
2. The abstract should be proofread for grammar, clarity, and coherence to
effectively communicate the intended information to the reader.
3. The introduction should not overly focus on the adverse effects of mental health
problems, and it should provide a more comprehensive description of the methodology.

Experimental design

4. In the literature section, improve the expression and provide transitions and
commentary between the application status of the Fireworks Algorithm and the K￾means algorithm.
5. Clearly state the specific research objectives and highlight the contribution to
the existing body of knowledge on mental health assessment.
6. Emphasize the importance of adopting a comprehensive psychological
perspective in mental health assessment, rather than solely relying on physical
indicators. Additionally, clarify how the new assessment indicators enrich the existing
psychological assessment indicator system and address the limitations of previous
indicators.

Validity of the findings

7. Provide a concise overview of the Fireworks Algorithm and its advantages in
addressing local optimization problems related to the K-means clustering algorithm.
8. In the conclusion section, discuss the potential practical significance and
applications of the findings in the field of mental health assessment for college students.

---

## Round 0.2 · accepted · Accept

Both reviewers have confirmed that the authors have addressed all of the reviewers' comments.

Reviewer 1 ·

Basic reporting

This article Assesse the college students' mental health status based on temporal perception and hybrid clustering algorithm under the impact of public health events, overall the article seems fine and contribute technically

Experimental design

Experimental design are improved in the revised version of the manuscript

Validity of the findings

I'm satisfied with the finding of the manuscript

Additional comments

I have no more comments and suggestions

Reviewer 2 ·

Basic reporting

The overall reporting of paper has improved

Experimental design

The experimental changes made are good; all the comments seem to be addressed.

Validity of the findings

The paper is overall good now and can be accepted